# Minimizing Polarization and Disagreement in Social Networks via Link Recommendation

**Liwang Zhu, Qi Bao, and Zhongzhi Zhang**[*]
Shanghai Key Lab of Intelligent Information Processing, Fudan University, Shanghai, China
School of Computer Science, Fudan University, Shanghai 200433, China
{19210240147, 20110240002, zhangzz}@fudan.edu.cn

## Abstract

Individual's opinions are fundamentally shaped and evolved by their interactions with other people, and social phenomena such as disagreement and polarization are now tightly woven into daily life. The quantification and optimization of these concepts have been the subject of much recent research behind a wealth of high-impact data mining applications. In particular, researchers have addressed the question of how such concepts can be optimized by influencing the opinion of a small number of individuals or by designing the network from scratch.

Here, rather than a "design-from-scratch" approach or altering the initial opinion, we study the optimization problem of recommending $k$ new links to minimize the sum of polarization and disagreement in a social network with $n$ nodes and $m$ edges. We show that our objective function of this combinatorial optimization problem is not submodular, although it is monotone. We propose a simple greedy algorithm with a constant-factor approximation that solves the problem in cubic running time, and we provide theoretical analysis of the approximation guarantee for the algorithm. To overcome the computation challenge for large networks, we also provide a fast algorithm with computation complexity $\widetilde{O}(mk\epsilon^{-2})$ for any $\epsilon > 0$, where the $\widetilde{O}(\cdot)$ notation suppresses the $\mathrm{poly}(\log n)$ factors. Extensive experiments on real datasets demonstrate both the efficiency and effectiveness of our algorithms.

## 1 Introduction

Social networks and social media play a prominent part in the propagation, evolution, and formulation of opinions [32], leading to fundamental changes in how humans share and shape opinions. Particularly, in recent years, the tremendous prevalence of online social media platforms produces various social phenomena, such as polarization and disagreement. The identification [48] and optimization [50] of these quantities are fundamental tasks behind a myriad of high-impact data mining applications, and thus have received considerable attention.

We focus on the phenomena of disagreement and polarization. Disagreement [39, 16] characterizes how much acquaintances disagree in their opinions, globally across the network. Polarization [39, 36, 16] measures how equilibrium expressed opinions deviate from their average. However, existing recommender systems, trained on real-data, with the goal to increase user engagement may stop the user from being exposed to diverse opinions and naturally end up creating "echo-chambers". In other words, the recommended links minimizing disagreement may lead to greater polarization [39] since connections between users with similar mindsets are preferred for such systems. Yet, exposure to diverse content is necessary to obtain a complete picture about a topic [27]. Thus, there is a need

---

[*]Corresponding author.

35th Conference on Neural Information Processing Systems (NeurIPS 2021).

for a radically different approach to suggest links that decrease both disagreement and polarization, which motivates our work.

In this paper, we address the following optimization problem: given a social network with $n$ nodes and $m$ edges, and an integer $k$, how to strategically recommend $k$ links to the individuals so that the polarization and disagreement is minimized. We show that our objective function of this combinatorial optimization problem is not submodular, although it is monotone. To tackle the exponential computation complexity, we resort to greedy algorithm extended for non-submodular function [5] by iteratively adding the most promising edges. We propose a simple greedy algorithm with a constant-factor approximation that solves the problem in cubic running time. To confront the computation challenge for large networks, we also provide a fast algorithm with computation complexity $\widetilde{O}(mk\epsilon^{-2})$ for any $\epsilon > 0$, where $\epsilon > 0$ is the error parameter and the $\widetilde{O}(\cdot)$ notation suppresses the $\mathrm{poly}(\log n)$ factors. We confirm our theoretical and algorithmic performance by executing extensive experiments on various real networks, which show that our algorithms are efficient and effective, outperforming several other strategies of creating edges.

Omitted proofs and implementation are provided as supplementary material.

## 2   Related work

We review the related literature from the following three perspectives, including modeling opinion dynamics, optimization problems in opinion dynamics, and link recommendation strategies.

**Modeling opinion dynamics.** Opinion dynamics has been the subject of intense recent research to model social learning processes in various disciplines [29, 19, 4]. These models capture the mechanisms and factors influencing opinion formulation, shedding light on understanding the whole process of opinion shaping and diverse phenomena taking place in social media. In the past decades, numerous relevant models have been proposed [40], among which the Friedkin-Johnsen (FJ) model [20] is one of the most popular models. After its establishment, the FJ model has been extended in a variety of ways [29, 28, 47, 13], by incorporating different factors affecting opinion dynamics, such as peer pressure [42], susceptibility to persuasion [1], and opinion leader [25]. Under the formalism of the FJ model or its variants, some relevant quantities, properties and explanations have been broadly studied, including the equilibrium expressed opinions [18, 6], sufficient condition for the stability [41], the average initial opinion [18], interpretations [24, 6], and so on.

**Optimization problems in opinion dynamics.** Recently, several optimization problems related to opinion dynamics have been formulated and studied for different objectives. For example, a long line of work has been devoted to maximizing the overall opinion by using different strategies, such as identifying a fixed number of individuals and setting their expressed opinions to 1 [25], changing agent's internal opinions [49, 47], as well as modifying individuals' susceptibility to persuasion [1, 8]. [46] studies the problem of allocating seed users to opposing campaigns with a goal to maximize the expected number of users who are co-exposed to both campaigns. In additon, [23] studies the problem of balancing the information exposure. These studies have far-reaching implications in product marketing, public health campaigns, and political candidates. Another major and increasingly important focus of research is optimizing some social phenomena, such as maximizing the diversity [35, 37], minimizing conflict [12], disagreement [21], and polarization [36]. A recent work considers the problem of minimizing the sum of polarization and disagreement by changing the underlying network topology [39]. Different from the "design-from-scratch" approach, we instead concentrate on edge recommendation, a practical incremental approach that suggests modifications to an existing network to achiever our goal. Moreover, our proposed algorithms differ significantly from existing ones, in spite of the fact that our objective function is identical to that in [39].

**Link recommendation strategies.** Our problem aims to optimize an objective via link recommendation. Here, we review some instances of related edge recommendation problems. [15, 43, 17] improve the centrality of a node and [3] strategically fight opinion control in social networks. As for opinion dynamics, the operation of creating edges is actually convenient and practical, which is equivalent to making friendship between individuals. In [22], the edge addition strategy is explored on the endorsement graph, with an aim to reduce controversy, and it also takes account of the acceptable probability of the recommended edge. In [6], addition of edges is discussed in order to reduce the social cost, namely the lack of agreement. Finally, [12] exploits link recommendation strategy to reduce the risk of conflict, [27] considers link insertions to reduce structural bias which may trap a

reader in a "polarized" bubble with no access to other opinions. Our work is motivated in part by these studies, while we address a new optimization problem, and present two new algorithms, with the faster one being nearly linear.

## 3  Preliminaries

In this section, we give a brief introduction to some essential concepts and tools, in order to facilitate the description of the studied problem and its related greedy algorithms.

**Graph and Related Matrix.** Let $\mathcal{G} = (V, E)$ be a connected undirected simple graph (network) with $|V| = n$ nodes and $|E| = m$ edges. The adjacency relation of all nodes in $\mathcal{G}$ is encoded in its adjacency matrix $\boldsymbol{A} = (a_{ij})_{n \times n}$. If nodes $i$ and $j$ are adjacent by an edge $e$, then $a_{ij} = a_{ji} = 1$; $a_{ij} = a_{ji} = 0$ otherwise. Let $N_i$ be the set of neighbours of node $i$ satisfying $N_i = \{j | \{i, j\} \in E\}$. Then, the degree $d_i$ of node $i$ is $d_i = \sum_{j=1}^{n} a_{ij} = \sum_{j \in N_i} a_{ij}$, and the diagonal degree matrix of $\mathcal{G}$ is defined as $\boldsymbol{D} = \mathrm{diag}(d_1, d_2, \ldots, d_n)$.

The Laplacian matrix of $\mathcal{G}$ is defined to be $\boldsymbol{L} = \boldsymbol{D} - \boldsymbol{A}$. There is also an alternative construction of $\boldsymbol{L}$ by using the incidence matrix $\boldsymbol{B} \in \mathbb{R}^{|E| \times |V|}$, an $m \times n$ signed edge-node incidence matrix. The element $b_{ev}$, $e \in E$ and $v \in V$, of $\boldsymbol{B}$ is defined as follows: $b_{ev} = 1$ if node $v$ is the head of edge $e$, $b_{ev} = -1$ if node $v$ is the tail of edge $e$, and $b_{ev} = 0$ otherwise. For an edge $e \in E$ with two end nodes $i$ and $j$, the row vector of $\boldsymbol{B}$ corresponding to $e$ can be written as $\boldsymbol{b}_{ij} \triangleq \boldsymbol{b}_e = \boldsymbol{e}_i - \boldsymbol{e}_j$ where $\boldsymbol{e}_i$ denotes the $i$-th standard basis vector of appropriate dimension. Then the Laplacian matrix $\boldsymbol{L}$ of $\mathcal{G}$ can also be represented as $\boldsymbol{L} = \boldsymbol{B}^{\top} \boldsymbol{B}$, which indicates that $\boldsymbol{L}$ is symmetric and positive semidefinite. Moreover, the Laplacian matrix $\boldsymbol{L}$ of a connected graph $\mathcal{G}$ has a unique zero eigenvalue.

Let $0 < \lambda_1 \leq \lambda_2 \leq \cdots \leq \lambda_{n-1}$ be the nonzero eigenvalues of $\boldsymbol{L}$ of a connected graph $\mathcal{G}$, and $\boldsymbol{u}_i$ be the corresponding orthonormal eigenvectors. Then, $\boldsymbol{L}$ has an eigendecomposition of form $\boldsymbol{L} = \boldsymbol{U} \Lambda \boldsymbol{U}^{\top} = \sum_{i=1}^{n-1} \lambda_i \boldsymbol{u}_i \boldsymbol{u}_i^{\top}$ where $\Lambda = \mathrm{diag}(0, \lambda_1, \lambda_2, .., \lambda_{n-1})$ and $\boldsymbol{u}_i$ is the $i$-th column of matrix $\boldsymbol{U}$. Let $\lambda_{\max}$ and $\lambda_{\min}$ be, respectively, the maximum and nonzero minimum eigenvalue of $\boldsymbol{L}$. Then, $\lambda_{\max} = \lambda_{n-1} \leq n$ [45], and $\lambda_{\min} = \lambda_1 \geq 1/n^2$ [34].

The forest matrix of graph $\mathcal{G}$ is defined as $\boldsymbol{\Omega} = (\boldsymbol{I} + \boldsymbol{L})^{-1} = (\omega_{ij})_{n \times n}$ [26, 11]. Matrix $\boldsymbol{\Omega}$ is a doubly stochastic matrix [9, 10], satisfying $\boldsymbol{\Omega} \mathbf{1} = \mathbf{1}$ and $\mathbf{1}^{\top} \boldsymbol{\Omega} = \mathbf{1}^{\top}$ where $\mathbf{1}$ denotes the all-ones vector. For a pair of nodes $i$ and $j$, their forest distance $r_{ij}$ is defined as $r_{ij} \triangleq \boldsymbol{b}_{ij}^{\top} \boldsymbol{\Omega} \boldsymbol{b}_{ij}$, obeying relation $0 \leq r_{ij} \leq 2$.

**Greedy Algorithms for Non-Submodular Functions.** For submodular maximization problems, the greedy algorithm has become a prevalent choice for solving them with a guaranteed $(1 - 1/e)$ approximation ratio. However, there are a large class of important set functions optimization in network topology design fail to be submodular. To alleviate this issue, a generalized approximation ratio has been proved and further provides a performance guarantee for non-submodular functions based on two quantities: submodularity ratio $\gamma$ and curvature $\alpha$, which characterize how close these functions are from being sub- or supermodular [5].

For consistency, we first give the definitions of submodular ratio and curvature below. We use $\rho_T(W) \overset{\text{def}}{=} f(T \cup W) - f(W)$ to denote the marginal benefit of the set $T \subseteq X$ with respect to the set $W \subseteq X$.

**Definition 3.1 (Submodular Ratio [5])** *The submodular ratio of a nonnegative set function $f$ is the largest $\gamma \in \mathbb{R}_+$ such that for any subset $W, T \subseteq X$, $\sum_{i \in W \setminus T} \rho_i(T) \geq \gamma \rho_W(T)$.*

**Definition 3.2 (Curvature [5])** *The curvature of a nonnegative set function $f$ is the smallest $\alpha \in \mathbb{R}_+$ such that for any subset $W, T \subseteq X$ and any element $j \in T \setminus W$, it follows that $\rho_j(T \setminus j \cup W) \geq (1 - \alpha) \rho_j(T \setminus j)$.*

Notice that for a non-decreasing function $f$, it follows that its submodular ratio $\gamma \in [0, 1]$ and $\gamma = 1$ if and only if $f$ is a submodular function; and its curvature $\alpha \in [0, 1]$ and $\alpha = 0$ if and only if $f$ is a supermodular function. Given those two quantities, the greedy algorithm enjoys a tight approximation guarantee of $\frac{1}{\alpha}(1 - e^{-\alpha\gamma})$ for a larger class of optimization problems [5].

## 4 Problem Formulation

In this section, we briefly discuss the FJ model of opinion formation, as well as the definitions and measures for disagreement and polarization. Then we formally introduce the problem for optimizing the polarization-disagreement index in a social network.

**Friedkin-Johnsen Model.** The FJ model is one of the most popular models for opinion dynamics [20]. In the FJ model, each node $i \in V$ has two opinions: internal (or innate) opinion and the expressed opinion, both in the interval $[0, 1]$. The internal opinion for each node, denoted by $s_i$, remains unchanged. Let $z_i(t)$ be the expressed opinion of node $i$ at time $t$, which evolves as

$$z_i(t+1) = \frac{s_i + \sum_{j \in N_i} a_{ij} z_j(t)}{1 + \sum_{j \in N_i} a_{ij}}. \tag{1}$$

Let $s = (s_1, s_2, \ldots, s_n)^\top$ be the initial opinion vector. Let $z = (z_1, z_2, \ldots, z_n)^\top$ be the equilibrium expressed opinion vector, which can be represented as $z = \Omega s$ [6].

In the FJ model, individuals interact with their acquaintances and exchange opinions whereas the opinions of nodes often do not reach consensus, leading to disagreement, polarization and other important phenomena, which have been the subject of many recent works. In this paper, we study the disagreement and polarization using definitions given in [39].

For a graph $\mathcal{G} = (V, E)$ with expressed opinion vector $z$, its disagreement $D(\mathcal{G})$ is defined as $D(\mathcal{G}) = \sum_{(i,j) \in E} (z_i - z_j)^2$. Let $\bar{z}$ be the mean-centered equilibrium vector given by $\bar{z} = z - \frac{z^\top 1}{n} 1$. Then the polarization $P(\mathcal{G})$ is defined as: $P(\mathcal{G}) = \sum_{i \in V} \bar{z}_i^2 = \bar{z}^\top \bar{z}$. We further introduce the polarization-disagreement (P-D) index, i.e., the objective we are concerned with. For a graph $\mathcal{G} = (V, E)$ with expressed opinion vector $z$, the polarization-disagreement index $\mathcal{I}(\mathcal{G})$ is the sum of the polarization $P(\mathcal{G})$ and disagreement $D(\mathcal{G}) : \mathcal{I}(\mathcal{G}) = P(\mathcal{G}) + D(\mathcal{G})$. Convenient matrix-vector expressions for the above quantities are provided by the following proposition [12, 39, 48].

**Proposition 1** $D(\mathcal{G})$, $P(\mathcal{G})$ and $\mathcal{I}(\mathcal{G})$ can be conveniently expressed in terms of quadratic forms as

$$D(\mathcal{G}) = z^\top L z = \bar{z}^\top L \bar{z} = s^\top \Omega L \Omega s,$$
$$P(\mathcal{G}) = \bar{z}^\top \bar{z} = \bar{s}^\top \Omega^2 \bar{s},$$
$$\mathcal{I}(\mathcal{G}) = \bar{z}^\top L \bar{z} + \bar{z}^\top \bar{z} = \bar{s}^\top \Omega \bar{s}.$$

**Remark 1** *In this paper, we will generally assume that the internal opinions $s$ are mean-centered, that is, $\bar{s} = s$. Note that in such case, $z$ will also be mean-centered. In fact, as will be shown in supplementary material, whether the opinions $s$ are mean-centered or not, the edges which we select to augment the network are the same since the variation of our objective remains unchanged.*

**Formulation of Problem.** Our problem is based on the following fact. For a connected undirected network $\mathcal{G}(V, E)$, if we augment the network by adding the edges in set $T$, which is a subset of the candidate edge set $E_C$ consisting of specified nonexistent edges, the P-D index of the new graph will not increase. In our subsequent analysis, we will use the following notation to improve readability of our lemmas. We use $\mathcal{G} + T$ to denote the graph augmented by adding the edges in $T$, that is $\mathcal{G} + T = (V, E \cup T)$. To evaluate the variation of the P-D index when adding edges, we define an objective function on the edge set $f : 2^{E_C} \to \mathbb{R}_+$ as

$$f(T) = \mathcal{I}(\mathcal{G}) - \mathcal{I}(\mathcal{G} + T), \tag{2}$$

in the sequel, we will use $f(e)$ and $\mathcal{G} + e$, respectively, to denote $f(\{e\})$ and $\mathcal{G} + \{e\}$ for simplicity. The variation of the P-D index under the perturbation of a single edge can be expressed by the following lemma.

**Lemma 4.1** *For a candidate edge $e \in E_C$ connecting node $u$ and $v$ with row vector $b_e = e_u - e_v$, one obtains*

$$f(e) = \frac{s^\top \Omega b_e b_e^\top \Omega s}{1 + b_e^\top \Omega b_e} = \frac{(z_u - z_v)^2}{1 + r_{uv}} \geq 0. \tag{3}$$

Lemma 4.1 indicates that the addition of any nonexisting edge can reduce the P-D index. Then the following problem arises naturally: given a candidate edge set $E_C$, how to optimally select a

subset $T$ of $E_C$ subject to a cardinality constraint, so that the polarization-disagreement index of the new graph is minimized. Such an optimization problem has been the subject of many recent papers [21, 39]. Yet we focus on edge recommendation rather than global structural results due to the following reasons: in practice, graph edits can typically be made only in small amounts, either because of budget constraints, or because of practical considerations. A similar idea was previously used in [12]. We now give a mathematical formulation of Problem 1 in a formal way as follows.

**Problem 1** *Given a connected graph $\mathcal{G} = (V, E)$, an integer $k$, a candidate edge set $E_C$ consisting of nonexistent edges, find a subset $T \subset E_C$ with $|T| = k$, and add them to $\mathcal{G}$ forming a new graph $\mathcal{G} + T = (V, E \cup T)$, so that the polarization-disagreement index $\mathcal{I}(\mathcal{G} + T)$ is minimized. That is:*

$$\underset{T \subset E_C, \, |T| = k}{\text{minimize}} \quad \mathcal{I}(\mathcal{G} + T). \tag{4}$$

## 5  SPGREEDY: Simple Greedy Algorithm

In this section, we first study the characterization of our problem, and then propose a simple greedy algorithm, followed by some analysis in terms of the approximation guarantee.

**Problem Characterization.** We start with detailing the theoretic challenges of Problem 1. The complexity of Problem 1 comes along two dimensions—searching for the best edge subset delivering the maximum of the objective, and assessing the impact of a given subset of edges upon the objective. The former is inherently a combinatorial problem, which is computationally infeasible to solve in a naïve brute-force manner while the latter involves cubic-time matrix inversion. To be specific, for each augmented edge set $T$ coming from $\binom{|E_C|}{k}$ possible subsets, we need to compute the P-D index of the resultant graph, yielding an exponential complexity $O\big(\binom{|E_C|}{k} \cdot n^3\big)$.

To tackle the exponential complexity, we resort to a greedy heuristic. Firstly, the objective function is monotone respect to the edge set $T$, as we mentioned in Lemma 4.1. But, we show that, unlike in the case in other standard optimization settings, neither submodularity nor supermodularity holds for our objective function. We present above results in the following theorem.

**Theorem 5.1** *The function $f(T)$ defined above is a non-submodular monotonically increasing function of the edge set $T$.*

Thanks to the empirical success of applying greedy strategy on a significantly larger class of non-submodular functions [5], in spite of the non-submodularity of our objective $f(T)$, the greedy algorithm can still become a proper choice for solving Problem 1, the details of which will be discussed in the next subsection.

**Simple Greedy Algorithm.** Our simple greedy algorithm, denoted as SPGREEDY, exploits the performance guarantee for non-submodular functions. The general approach is as follows. We will first assess candidate edges with respect to how much their addition to the network decreases the term $\mathcal{I}(\mathcal{G})$, and then iteratively add the most promising edges to the network until budget is reached.

The augmented edge set $T$ is initialized with an empty set. Then $k$ edges are iteratively selected to the augmented edge set from set $E_C \setminus T$. At each iteration of the greedy algorithm, the edge $e$ in candidate set is chosen that gives the largest marginal gain $f(e)$. The algorithm stops until $k$ edges are selected to be added to $T$. The naïve greedy algorithm takes time $O(k|E_C|n^3)$, which is computationally intractable even for small-size networks. Actually, as shown in the proof of Lemma 4.1, with $\mathbf{\Omega}$ already computed, we can view the addition of a single edge $e$ as a rank-1 update to the original matrix $\mathbf{\Omega}$, which can be calculated in time $O(n^2)$ by using Sherman-Morrison formula [38], instead of inverting the matrix again in $O(n^3)$ in each loop.

The above analysis leads to a simple greedy algorithm SPGREEDY$(\mathcal{G}, E_C, k, \boldsymbol{s})$, which is outlined in Algorithm 1. To begin with, this algorithm requires $O(n^3)$ time to compute the inverse of $\boldsymbol{I} + \boldsymbol{L}$, and then it performs in $k$ rounds, with each round mainly including two steps: computing $f(e)$ (Line 4) in $O(|E_C|n^2)$ time, and updating $\mathbf{\Omega}$ (Line 8) in $O(n^2)$ time. Thus, the total running time of Algorithm 1 is $O(n^3 + k|E_C|n^2)$, which is much faster than the naïve algorithm.

**Approximation Guarantee.** As we mentioned before, the performance of the greedy algorithm for non-submodular function can be evaluated by its submodularity ratio $\gamma$ and curvature $\alpha$. Hence, we derive the bounds for these two quantities of our Problem 1.

---

**Algorithm 1:** SPGREEDY($\mathcal{G}, E_C, k, \boldsymbol{s}$)

---

**Input** : A connected graph $\mathcal{G}$; a candidate edge set $E_C$; an integer $k \leq |E_C|$; an initial opinion
vector $\boldsymbol{s}$
**Output** : A subset of $T \subset E_C$ and $|T| = k$

**1** Initialize solution $T = \emptyset$
**2** Compute $\boldsymbol{\Omega}$
**3 for** $i = 1$ *to* $k$ **do**
**4**  $\quad$ Compute $f(e)$ for each $e \in E_C \setminus T$
**5**  $\quad$ Select $e_i$ s.t. $e_i \leftarrow \arg\max_{e \in E_C \setminus T} f(e)$
**6**  $\quad$ Update solution $T \leftarrow T \cup \{e_i\}$
**7**  $\quad$ Update the graph $\mathcal{G} \leftarrow \mathcal{G}(V, E \cup \{e_i\})$
**8**  $\quad$ Update $\boldsymbol{\Omega} \leftarrow \boldsymbol{\Omega} - \frac{\boldsymbol{\Omega} \boldsymbol{b}_e \boldsymbol{b}_e^\top \boldsymbol{\Omega}}{1 + \boldsymbol{b}_e^\top \boldsymbol{\Omega} \boldsymbol{b}_e}$

**9 return** $T$

---

**Lemma 5.1** *Let $\lambda_1(\boldsymbol{L})$ be the smallest non-zero eigenvalue of Laplacian matrix $\boldsymbol{L}$ for graph $\mathcal{G}$, and let $\lambda_{n-1}(\boldsymbol{L}_{E_C})$ be the largest eigenvalue of Laplacian matrix $\boldsymbol{L}_{E_C}$ for the augmented graph $\mathcal{G} + E_C$. Then, the submodularity ratio $\gamma$ of set function $f(T) = \mathcal{I}(\mathcal{G}) - \mathcal{I}(\mathcal{G} + T)$ is bounded by $1 > \gamma \geq \left(\frac{1 + \lambda_1(\boldsymbol{L})}{1 + \lambda_{n-1}(\boldsymbol{L}_{E_C})}\right)^2 > 0$, and its curvature $\alpha$ is bounded by $0 < \alpha \leq 1 - \left(\frac{1 + \lambda_1(\boldsymbol{L})}{1 + \lambda_{n-1}(\boldsymbol{L}_{E_C})}\right)^2 < 1$.*

Lemma 5.1, together with the approximation guarantee stated before, yields a performance analysis for the greedy algorithm. As will be shown in the Experiment Section, the greedy algorithm has been shown to perform very close to the optimal solutions for our problem in the experimental aspect.

## 6 FASTGREEDY: Fast Greedy Algorithm

Compared with the naïve algorithm, computation time of Algorithm 1 is significantly reduced. However, it is still computationally unacceptable when employed on large networks with millions of vertices. In this section, we address this challenge by presenting an efficient approximation algorithm, solving the problem in time $\widetilde{O}(mk\epsilon^{-2})$.

The core step for solving Problem 1 is to calculate the impact of each edge upon the objective i.e., $f(e)$. According to (3), to evaluate $f(e)$, we need to estimate two terms $\boldsymbol{s}^\top \boldsymbol{\Omega} \boldsymbol{b}_e \boldsymbol{b}_e^\top \boldsymbol{\Omega} \boldsymbol{s}$ and $\boldsymbol{b}_e^\top \boldsymbol{\Omega} \boldsymbol{b}_e$. We will show how to approximate these two quantities in nearly linear time using Johnson-Lindenstrauss (JL) Lemma [30, 2] and Fast SDDM (symmetric, diagonally-dominant M-matrix) Solvers[44]. Moreover, the results returned by our algorithm are demonstrated to provide proper approximations to $f(e)$.

**Lemma 6.1 (JL Lemma [30])** *Given fixed vectors $\boldsymbol{v}_1, \boldsymbol{v}_2, \cdots, \boldsymbol{v}_n \in \mathbb{R}^d$ and a real number $\epsilon > 0$. Let $p$ be a positive integer such that $p \geq 24 \log n / \epsilon^2$ and $\boldsymbol{R}_{p \times d}$ a random matrix with each entry being $1/\sqrt{p}$ or $-1/\sqrt{p}$ with identical probability. Then, with probability at least $1 - 1/n$, the following statement holds for any pair of $i$ and $j$, $1 \leq i, j \leq n$:*

$$(1 - \epsilon)\|\boldsymbol{v}_i - \boldsymbol{v}_j\|^2 \leq \|\boldsymbol{R}\boldsymbol{v}_i - \boldsymbol{R}\boldsymbol{v}_j\|^2 \leq (1 + \epsilon)\|\boldsymbol{v}_i - \boldsymbol{v}_j\|^2.$$

**Lemma 6.2 (Fast SDDM Solver [44])** *There is a nearly linear time solver $\boldsymbol{y} = \text{SOLVE}(\boldsymbol{S}, \boldsymbol{b}, \epsilon)$ which takes an SDDM matrix $\boldsymbol{S}_{n \times n}$ with $m$ nonzero entries, a vector $\boldsymbol{b} \in \mathbb{R}^n$, and an error parameter $\delta > 0$, and returns a vector $\boldsymbol{x} \in \mathbb{R}^n$ satisfying $\|\boldsymbol{y} - \boldsymbol{S}^{-1}\boldsymbol{b}\|_{\boldsymbol{S}} \leq \delta \|\boldsymbol{S}^{-1}\boldsymbol{b}\|_{\boldsymbol{S}}$ with high probability, where $\|\boldsymbol{y}\|_{\boldsymbol{S}} \stackrel{\text{def}}{=} \sqrt{\boldsymbol{y}^\top \boldsymbol{S} \boldsymbol{y}}$. The solver runs in expected time $\widetilde{O}(m)$.*

We first approximate the term $\boldsymbol{b}_e^\top \boldsymbol{\Omega} \boldsymbol{b}_e$ in the denominator of (3), which can be written as $\boldsymbol{b}_e^\top \boldsymbol{\Omega} \boldsymbol{b}_e = \boldsymbol{b}_e^\top \boldsymbol{\Omega}(\boldsymbol{I} + \boldsymbol{L})\boldsymbol{\Omega} \boldsymbol{b}_e = \boldsymbol{b}_e^\top \boldsymbol{\Omega}\left(\boldsymbol{I} + \boldsymbol{B}^\top \boldsymbol{B}\right)\boldsymbol{\Omega} \boldsymbol{b}_e = \|\boldsymbol{\Omega} \boldsymbol{b}_e\|^2 + \|\boldsymbol{B}\boldsymbol{\Omega} \boldsymbol{b}_e\|^2$. In this way, we have reduced the estimation of the denominator of (3) to the calculation of the quadratic form of $\|\boldsymbol{B}\boldsymbol{\Omega} \boldsymbol{b}_e\|^2$ and $\|\boldsymbol{\Omega} \boldsymbol{b}_e\|^2$ in $\mathbb{R}^m$ and $\mathbb{R}^n$. To reduce the computation cost, we will apply the JL lemma to reduce the dimensions. Let $\boldsymbol{Q}_{p \times m}$ and $\boldsymbol{P}_{p \times n}$ be two random $\pm 1/\sqrt{p}$ matrices where $p$ will be decided later,

then we can simply project matrices $\boldsymbol{B\Omega}$ and $\boldsymbol{\Omega}$ onto low-dimensional subspace $\boldsymbol{QB\Omega}$ and $\boldsymbol{P\Omega}$. However, this still does not help to reduce the computation time, since direct computation of the above $\ell_2$ norm involves matrix inversion, leading a running time of $O(n^3)$. In order to avoid inverting matrix $\boldsymbol{I} + \boldsymbol{L}$, we will utilize the fast SDDM linear system solvers [44] to approximate the above two terms.

Let $\boldsymbol{X} = \boldsymbol{QB}$, $\overline{\boldsymbol{X}} = \boldsymbol{B\Omega}$, $\boldsymbol{X}' = \boldsymbol{Q}\overline{\boldsymbol{X}}$, let $\tilde{\boldsymbol{X}}_i = \text{SOLVE}(\boldsymbol{I} + \boldsymbol{L}, \boldsymbol{X}_i, \delta_1)$, satisfying $\|\tilde{\boldsymbol{X}}_i - \boldsymbol{X}'_i\|_{\boldsymbol{I}+\boldsymbol{L}} \le \delta_1 \|\boldsymbol{X}'_i\|_{\boldsymbol{I}+\boldsymbol{L}}$ with $\delta_1 \le \frac{\epsilon\sqrt{2(1-\epsilon/12)}}{32n(n+1)\sqrt{(1+\epsilon/12)(n+1)}}$. Making use of Lemmas 6.1 and 6.2, the term $\|\boldsymbol{QB\Omega b}_e\|^2$ can be efficiently approximated as stated in the following lemma.

**Lemma 6.3** *Given an undirected graph $\mathcal{G} = (V, E)$ with Laplacian matrix $\boldsymbol{L}$, a parameter $\epsilon \in (0, \frac{1}{2})$, then, the following relation holds:*

$$(1 - \frac{\epsilon}{12})^2 \|\overline{\boldsymbol{X}}\boldsymbol{b}_e\|^2 \le \|\tilde{\boldsymbol{X}}\boldsymbol{b}_e\|^2 \le (1 + \frac{\epsilon}{12})^2 \|\overline{\boldsymbol{X}}\boldsymbol{b}_e\|^2. \tag{5}$$

In a similar way, let $\boldsymbol{Y} = \boldsymbol{P}$, $\overline{\boldsymbol{Y}} = \boldsymbol{\Omega}$, $\boldsymbol{Y}' = \boldsymbol{P}\overline{\boldsymbol{Y}}$, let $\tilde{\boldsymbol{Y}}_i = \text{SOLVE}(\boldsymbol{I} + \boldsymbol{L}, \boldsymbol{Y}_i, \delta_2)$ with $\delta_2 \le \frac{\epsilon\sqrt{2(1-\epsilon/12)}}{32(n+1)\sqrt{(1+\epsilon/12)(n+1)}}$. Then, the following lemma holds, giving an efficient approximation to $\|\boldsymbol{Q\Omega b}_e\|^2$.

**Lemma 6.4** *Given an undirected graph $\mathcal{G} = (V, E)$ with Laplacian matrix $\boldsymbol{L}$, a parameter $\epsilon \in (0, \frac{1}{2})$, then, the following relation holds:*

$$(1 - \frac{\epsilon}{12})^2 \|\overline{\boldsymbol{Y}}\boldsymbol{b}_e\|^2 \le \|\tilde{\boldsymbol{Y}}\boldsymbol{b}_e\| \le (1 + \frac{\epsilon}{12})^2 \|\overline{\boldsymbol{Y}}\boldsymbol{b}_e\|^2. \tag{6}$$

Having Lemma 6.3, 6.4, the term $\boldsymbol{b}_e^\top \boldsymbol{\Omega b}_e$ can be efficiently approximated by $\|\tilde{\boldsymbol{X}}\boldsymbol{b}_e\|^2 + \|\tilde{\boldsymbol{Y}}\boldsymbol{b}_e\|^2$ satisfying

$$(1 - \frac{\epsilon}{3})\boldsymbol{b}_e^\top \boldsymbol{\Omega b}_e \le \|\tilde{\boldsymbol{X}}\boldsymbol{b}_e\|^2 + \|\tilde{\boldsymbol{Y}}\boldsymbol{b}_e\|^2 \le (1 + \frac{\epsilon}{3})\boldsymbol{b}_e^\top \boldsymbol{\Omega b}_e.$$

We continue to approximate the numerator of $f(e)$, which can be written as $\|\boldsymbol{\Omega s}\|^2_{\boldsymbol{b}_e \boldsymbol{b}_e^\top}$. We will use the fast SDDM matrix solver again to approximate this term.

**Lemma 6.5** *Given an undirected graph $\mathcal{G} = (V, E)$ with Laplacian matrix $\boldsymbol{L}$ and forest matrix $\boldsymbol{\Omega}$, a parameter $\epsilon \in (0, \frac{1}{2})$, and the internal opinion vector $\boldsymbol{s}$, let $\boldsymbol{q} = \text{SOLVE}(\boldsymbol{I} + \boldsymbol{L}, \boldsymbol{s}, \delta_3)$, where $\delta_3 \le \frac{\epsilon}{9n^2}$. Then, the following relation holds: $\left| \|\boldsymbol{q}\|^2_{\boldsymbol{b}_e \boldsymbol{b}_e^\top} - \|\boldsymbol{\Omega s}\|^2_{\boldsymbol{b}_e \boldsymbol{b}_e^\top} \right| \le \frac{\epsilon}{3}$.*

---

**Algorithm 2:** $\text{COMP}(\mathcal{G}, E_C, \boldsymbol{s}, \epsilon)$

---

**Input**    : A graph $\mathcal{G}$; a candidate edge set $E_C$; an initial opinion vector $\boldsymbol{s}$; a real number $0 \le \epsilon \le 1/2$

**Output**  : $\{(e, \hat{f}(e)|e \in E_C\}$

1  Set $\delta_1, \delta_2, \delta_3$ according to Lemmas 6.3, 6.4 and 6.5,
2  $p \leftarrow \lceil 24 \log n/(\frac{\epsilon}{12})^2 \rceil$
3  Generate random Gaussian matrices $\boldsymbol{P}_{p \times n}$, $\boldsymbol{Q}_{p \times m}$
4  Compute approximations $\boldsymbol{q}$ to $\boldsymbol{\Omega s}$, $\tilde{\boldsymbol{X}}$ to $\overline{\boldsymbol{X}} \stackrel{\text{def}}{=} \boldsymbol{B\Omega}$, and $\tilde{\boldsymbol{Y}}$ to $\overline{\boldsymbol{Y}} \stackrel{\text{def}}{=} \boldsymbol{\Omega}$
5  **for** $i = 1$ *to* $p$ **do**
6  $\quad$ $\boldsymbol{q} \leftarrow \text{SOLVE}(\boldsymbol{I} + \boldsymbol{L}, \boldsymbol{s}, \delta_3)$
7  $\quad$ $\tilde{\boldsymbol{X}}_i \leftarrow \text{SOLVE}(\boldsymbol{I} + \boldsymbol{L}, \boldsymbol{X}_i, \delta_1)$
8  $\quad$ $\tilde{\boldsymbol{Y}}_i \leftarrow \text{SOLVE}(\boldsymbol{I} + \boldsymbol{L}, \boldsymbol{Y}_i, \delta_2)$
9  **for** *each* $e \in E_C$ **do**
10 $\quad$ compute $\hat{f}(e) = \frac{\boldsymbol{b}_e^\top \boldsymbol{q} \boldsymbol{q}^\top \boldsymbol{b}_e}{1 + \|\tilde{\boldsymbol{X}}\boldsymbol{b}_e\|^2 + \|\tilde{\boldsymbol{Y}}\boldsymbol{b}_e\|^2}$
11 **return** $\{(e, \hat{f}(e)|e \in E_C\}$

---

Having considerably approximately computing each part of $f(e)$, we are now able to propose an algorithm COMP approximating $f(e)$ for every edge $e$ in the candidate set $E_C$ based on Lemmas 6.3, 6.4 and 6.5. The outline of algorithm COMP is shown in Algorithm 2, whose performance is given in Theorem 6.1.

**Theorem 6.1** *For $0 \le \epsilon \le 1/2$, the value $\hat{f}(e)$ returned by* COMP *satisfies* $|f(e) - \hat{f}(e)| \le \frac{4}{5}\epsilon$ *with high probability.*

Exploiting Algorithm 2 to approximate $f(e)$, we develop a fast greedy algorithm FASTGREEDY$(\mathcal{G}, E_C, \boldsymbol{s}, k, \epsilon)$ in Algorithm 3 to solve Problem 1. Overall, it follows the greedy strategy (Algorithm 1). In detail, Algorithm 3 performs $k$ rounds (Lines 2-6). In each round, it takes time $\widetilde{O}(m\epsilon^{-2})$ to call COMP to compute the approximation of the marginal gain $\hat{f}(e)$, then iteratively select the element with the highest impact score and update. Therefore, the time complexity of Algorithm 3 is $\widetilde{O}(mk\epsilon^{-2})$.

---

**Algorithm 3:** FASTGREEDY$(\mathcal{G}, E_C, \boldsymbol{s}, k, \epsilon)$

**Input** : A graph $\mathcal{G}$; a candidate edge set $E_C$; an initial opinion vector $\boldsymbol{s}$; an integer $k \le |E_C|$;
a real number $0 \le \epsilon \le 1/2$
**Output** : $T$: a subset of $E_C$ and $|T| = k$
1 Initialize solution $T = \emptyset$
2 **for** $i = 1$ *to* $k$ **do**
3 $\quad \{e, \hat{f}(e)|e \in E_C \setminus T\} \leftarrow$ COMP$(\mathcal{G}, E_C \setminus T, \boldsymbol{s}, \epsilon)$
4 $\quad$ Select $e_i$ s.t. $e_i \leftarrow \arg\max_{e \in E_C \setminus T} \hat{f}(e)$
5 $\quad$ Update solution $T \leftarrow T \cup \{e_i\}$
6 $\quad$ Update the graph $\mathcal{G} \leftarrow \mathcal{G}(V, E \cup \{e_i\})$
7 **return** $T$

---

## 7 Experiments

In this section, we present numerical results to evaluate the performance of our two greedy algorithms SPGREEDY and FASTGREEDY. To this end, extensive experiments are designed and executed on real networks of various types and scales to validate the effectiveness and efficiency of our algorithms.

**Datasets.** The studied realistic networks are representatively selected from various domains, which are publicly available in the KONECT [31] and SNAP [33]. For each network, we implement our experiments on its largest components. The characteristics of the largest components for all networks are summarized in Table 1.

**Machine and reproducibility.** All experiments were conducted on a machine equipped with 32G RAM and 4.2 GHz Intel i7-7700 CPU. All algorithms in our experiments are executed in Julia. In our algorithms, we use the SDDM solver SOLVE [44], the Julia implementation of which is available at https://github. com/danspielman/Laplacians.jl.

Table 1: Running time and results for each dataset with $k$=50.

| Network | $n$ | $m$ | Running Time (s) | | $\Delta\mathcal{I}(\mathcal{G})$ | | Ratio |
|---|---|---|---|---|---|---|---|
| | | | SPGREEDY | FASTGREEDY | SPGREEDY | FASTGREEDY | |
| GrQc | 4,158 | 13,422 | 7.83 | 11.13 | -4.9966 | -4.9774 | 0.9962 |
| USgrid | 4,941 | 6,594 | 10.31 | 10.70 | -7.3406 | -7.2720 | 0.9906 |
| Erdös992 | 5,094 | 7,515 | 10.91 | 8.98 | -6.1616 | -6.0984 | 0.9897 |
| Bcspwr10 | 5,300 | 8,271 | 12.03 | 11.57 | -6.0290 | -5.9820 | 0.9922 |
| Reality | 6,809 | 7,680 | 19.81 | 7.83 | -5.7861 | -5.7183 | 0.9883 |
| PagesGovernment | 7,057 | 89,429 | 21.10 | 27.27 | -2.7580 | -2.7484 | 0.9965 |
| WikiElec | 7,115 | 100,753 | 21.13 | 27.74 | -4.7478 | -4.7105 | 0.9921 |
| Dmela | 7,393 | 25,569 | 23.59 | 15.70 | -4.9245 | -4.8988 | 0.9948 |
| HepPh | 11,204 | 117,619 | 59.37 | 79.10 | -4.2728 | -4.2641 | 0.9980 |
| Anybeat | 12,645 | 49,132 | 81.86 | 44.41 | -5.4685 | -5.4180 | 0.9908 |
| PagesCompany | 14,113 | 52,126 | 106.40 | 62.88 | -5.2711 | -5.2554 | 0.9970 |
| CondMat | 21,363 | 91,286 | 319.86 | 101.95 | -4.5719 | -4.5659 | 0.9987 |
| Gplus | 23,628 | 39,194 | 416.24 | 42.99 | -5.3146 | -5.2672 | 0.9911 |
| Brightkite* | 56,739 | 212,945 | — | 274.97 | — | — | — |
| WikiTalk* | 92,117 | 360,767 | — | 397.34 | — | — | — |
| Douban* | 154,908 | 327,162 | — | 494.19 | — | — | — |
| Citeseer* | 227,320 | 814,134 | — | 1460.20 | — | — | — |
| TwitterFollows* | 404,719 | 713,319 | — | 1009.49 | — | — | — |
| FourSquare* | 639,014 | 3,214,986 | — | 3748.63 | — | — | — |
| IMDB* | 896,305 | 3,782,447 | — | 9076.97 | — | — | — |
| YoutubeSnap* | 1,134,890 | 2,987,624 | — | 6918.97 | — | — | — |

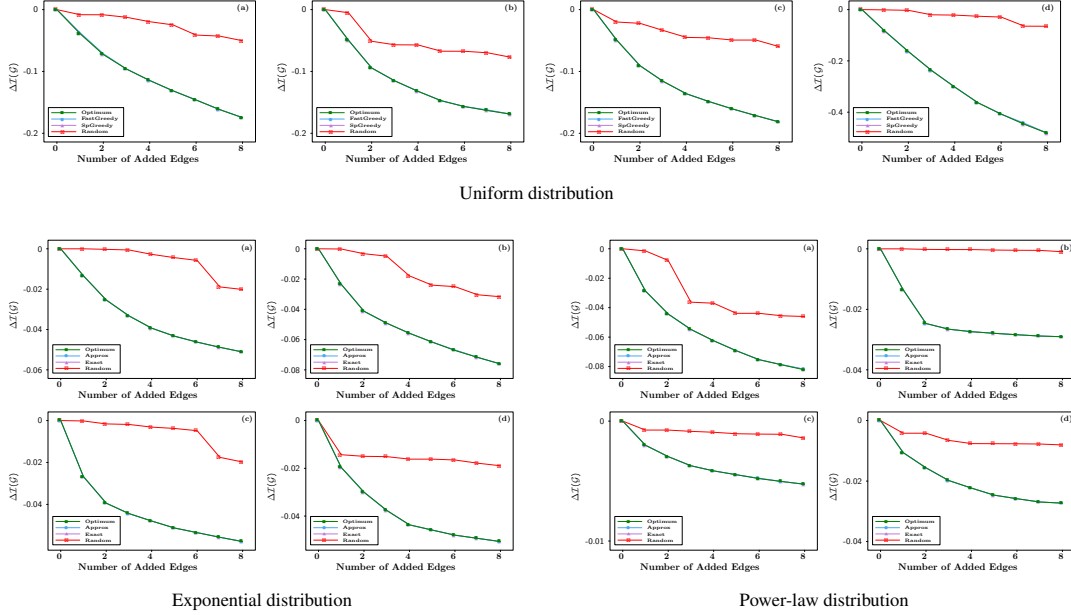

Uniform distribution

Exponential distribution                    Power-law distribution

Figure 1: Optimization results of four methods for edge addition on datasets (a) Karate, (b) Dolphins, (c) Netscience and (d) Diseasom with three initial opinion distributions for varying $k$.

**Methods.** The sets of $k$ edges are found with the following seven strategies. (1) *Optimum*: selecting edge set $|T| = k$ with minimum $\mathcal{I}(\mathcal{G} + T)$ by exhaustive search. (2) *Random*: selecting $k$ edges at random. (3) *Betweenness*: selecting top-$k$ edges with the highest betweenness [7]. (4) *DegProduct*: selecting top-$k$ edges with the largest product of two end-points degrees. (5) *DegSum*: selecting top-$k$ edges with the largest sum of two end-points degrees. (6)*FastGreedy*: selecting top-$k$ edges with maximum $f(e)$ returned by algorithm FASTGREEDY. (7) *SpGreedy*: selecting top-$k$ edges with maximum $f(e)$ returned by algorithm SPGREEDY.

**Opinions and evaluation metrics.** We investigate three different distributions for the initial opinions: uniform distribution, exponential distribution, and power-law distribution. For the uniform distribution, the initial opinion $\boldsymbol{s}_i$ of each node $i$ is distributed uniformly in the range of $[0, 1]$. For the latter two distributions, we use randht.py file in [14] to generate and normalize opinions to the range of $[0, 1]$ according to a exponential distribution and a power law with a given slope. Note that there is always a node with internal opinion 1 due to the normalization operation. The performance of all above-mentioned methods is evaluated by the impact of their selected edges on the drop of the P-D index $\Delta\mathcal{I}(\mathcal{G}) = \mathcal{I}(\mathcal{G} + T) - \mathcal{I}(\mathcal{G})$, with a smaller $\Delta\mathcal{I}(\mathcal{G})$ corresponding to an more effective method.

**Results.** We first evaluate the effectiveness of our algorithms, we execute experiments on four small realistic networks: Karate with 34 nodes and 78 edges, Dolphins with 62 nodes and 159 edges, Netscience with 379 nodes and 914 edges, and Diseasom with 516 nodes and 1188 edges. These networks are small, allowing us to compute the optimal set of edges. For each case, we add $k = 1, 2, \ldots, 8$ edges to augment the network from $|E_C| = 30$ candidate edges. For the approximation algorithm FASTGREEDY, we set $\epsilon = 0.3$. Note that a smaller $\epsilon$ corresponds to a stronger approximation ratio but less efficiency. As shown in Figures 1, for all three initial opinion distributions, the solutions returned by our two greedy algorithms and the optimum solution are almost the same so that the three curves are overlapped, demonstrating that our greedy algorithms perform much better than the theoretical guarantee. In addition, both of our algorithms are much better than those returned by the random scheme.

To further show the effectiveness of our algorithms, we proceed to compare the results of our methods with the baseline schemes on four relatively larger real-world networks: Yeast with 1458 nodes and 1948 edges, GridWorm with 3507 nodes and 6531 edges, Erdös992 with 5094 nodes and 7515 edges, and Reality with 6809 nodes and 7680 edges, for which it is almost impossible to get the optimal solutions by brute-force search. For each network, the performance of different methods for

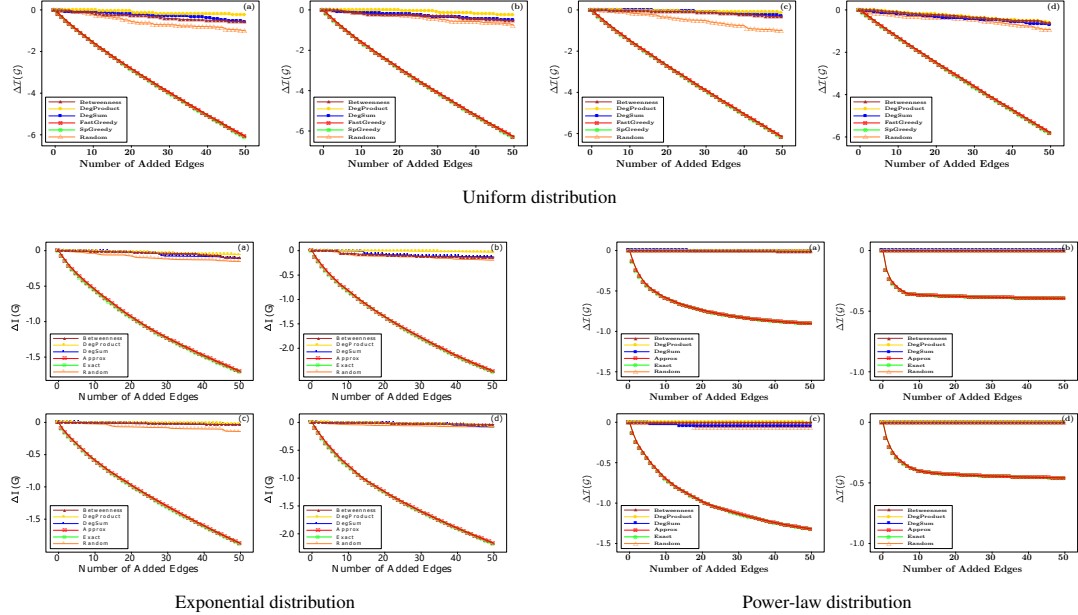

Figure 2: Optimization results for six methods of edge addition on datasets: (a) Yeast, (b) GridWorm, (c) Erdös992 and (d) Reality with three initial opinion distributions for varying $k$.

varying $k$ and $|E_C| = 10000$ candidate edges are displayed in Figures 2. We can see that SPGREEDY achieves the best performance as expected, and the proposed FASTGREEDY (a): is very close to the SPGREEDY method, and (b): consistently outperforms all four alternative methods.

In Table 1 we provide the results of running time and the drop of the P-D index $\Delta\mathcal{I}(\mathcal{G})$ returned by our two greedy algorithms. We observe that FASTGREEDY is significantly faster than SPGREEDY, especially for large networks, while both algorithms almost yield the same value on $\Delta\mathcal{I}(\mathcal{G})$. It is worth noting that SPGREEDY is not applicable to the last eight networks marked with "*" due to the limitations of time and memory. In comparison with SPGREEDY, FASTGREEDY approximately computes $\Delta\mathcal{I}(\mathcal{G})$ within several hours. Therefore, our algorithm FASTGREEDY achieves remarkable improvement in efficiency and is scalable to large networks with more than $10^6$ nodes.

## 8   Conclusions

In this paper, we considered the problem of minimizing the disagreement polarization index $\mathcal{I}(\mathcal{G})$ by strategically recommending $k$ links. This problem belongs to the class of discrete optimization that has found vast applications in various domains. We showed the objective function is monotone but non-submodular. We then presented two greedy algorithms to solve the optimization problem: the former returns a constant-factor approximation of the optimum solutions in time $O(n^3)$, while the latter runs in time $\widetilde{O}(mk\epsilon^{-2})$. On the theoretic side, we provided analysis of the approximation guarantee for these two algorithms. On the experimental aspect, we performed extensive experiments on real-life networks, which demonstrate that both of our algorithms lead to almost optimal solutions, and consistently outperform several alternative baseline heuristics. Particularly, our second algorithm could yield a good approximation solution quickly on networks with more than one million nodes within 2 hours, demonstrating excellent scalablity to massive networks.

## Acknowledgments and Disclosure of Funding

This work was supported by the National Key R & D Program of China (No. 2018YFB1305104), the National Natural Science Foundation of China (Nos. U20B2051 and 61872093), Shanghai Municipal Science and Technology Major Project (Nos. 2018SHZDZX01 and 2021SHZDZX03), ZJ Lab, and Shanghai Center for Brain Science and Brain-Inspired Technology.

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
