# Supplementary Material for: Minimizing Polarization and Disagreement in Social Networks via Link Recommendation

**Liwang Zhu, Qi Bao, and Zhongzhi Zhang**[*]
Shanghai Key Lab of Intelligent Information Processing, Fudan University, Shanghai, China
School of Computer Science, Fudan University, Shanghai 200433, China
{19210240147, 20110240002, zhangzz}@fudan.edu.cn

## 1  Proof of Lemma 4.1

**Proof.**  If we perturb the network with the addition of $e$, we obtain the new Laplacian $\boldsymbol{L} + \boldsymbol{b}_e \boldsymbol{b}_e^\top$. By Sherman-Morrison formula [1], we obtain

$$\left( \boldsymbol{I} + \boldsymbol{L} + \boldsymbol{b}_e \boldsymbol{b}_e^\top \right)^{-1} = (\boldsymbol{I} + \boldsymbol{L})^{-1} - \frac{\boldsymbol{\Omega} \boldsymbol{b}_e \boldsymbol{b}_e^\top \boldsymbol{\Omega}}{1 + \boldsymbol{b}_e^\top \boldsymbol{\Omega} \boldsymbol{b}_e}.$$

By the definitions of P-D index, $\mathcal{I}(\mathcal{G} + e) = \boldsymbol{s}^\top (\boldsymbol{I} + \boldsymbol{L} + \boldsymbol{b}_e \boldsymbol{b}_e^\top)^{-1} \boldsymbol{s}$. We can immediately obtain $f(e) = \mathcal{I}(\mathcal{G}) - \mathcal{I}(\mathcal{G} + e) = \frac{(\boldsymbol{s}^\top \boldsymbol{\Omega} \boldsymbol{b}_e)^2}{1 + \boldsymbol{b}_e^\top \boldsymbol{\Omega} \boldsymbol{b}_e}$. Since the term $(\boldsymbol{s}^\top \boldsymbol{\Omega} \boldsymbol{b}_e)^2 = (\boldsymbol{z}_u - \boldsymbol{z}_v)^2$ is nonnegative, together with the fact that $0 \le \boldsymbol{b}_e^\top \boldsymbol{\Omega} \boldsymbol{b}_e \le 2$, we can conclude $f(e) \ge 0$ consequently. □

## 2  Proof of Remark 1

**Proof.**  When the opinions $\boldsymbol{s}$ are mean-centered, corresponding variation of the objective could be expressed as

$$
\begin{aligned}
f(e) &= \overline{\boldsymbol{s}}^\top \boldsymbol{\Omega} \boldsymbol{b}_e \boldsymbol{b}_e^\top \boldsymbol{\Omega} \overline{\boldsymbol{s}} = \left( (\boldsymbol{s} - \frac{\boldsymbol{s}^\top \boldsymbol{1}}{n} \boldsymbol{1})^\top \boldsymbol{\Omega} \boldsymbol{b}_e \right)^2 \\
&= \left( \boldsymbol{s}^\top \boldsymbol{\Omega} \boldsymbol{b}_e - \frac{\boldsymbol{s}^\top \boldsymbol{1}}{n} \boldsymbol{1}^\top \boldsymbol{\Omega} \boldsymbol{b}_e \right)^2 = \boldsymbol{s}^\top \boldsymbol{\Omega} \boldsymbol{b}_e \boldsymbol{b}_e^\top \boldsymbol{\Omega} \boldsymbol{s},
\end{aligned}
$$

where the last equality is obtained by the fact that $\boldsymbol{1}^\top \boldsymbol{\Omega} \boldsymbol{b}_e = \boldsymbol{1}^\top \boldsymbol{b}_e = 0$.

Thus, under the perturbation of the network with a single edge $e$, it holds that whether the opinions $\boldsymbol{s}$ are mean-centered or not, the variation of our objective i.e. $f(e)$ are the same. The above results complete the proof. □

## 3  Proof of Lemma 5.1

**Proof.**  Note, for two matrices $\boldsymbol{A}$ and $\boldsymbol{B}$, we write $\boldsymbol{A} \preceq \boldsymbol{B}$ to denote that $\boldsymbol{B} - \boldsymbol{A}$ is positive semidefinite. We use $(\boldsymbol{I} + \boldsymbol{L})_T^{-1}$ to denote the forest matrix associated with graph $\mathcal{G} + T$.

Let $E_C$ be the candidate set, and let $T, W$ be any two subsets of $E_C$. To begin with, we first derive a lower and an upper bound, respectively, for the marginal benefit function $\rho_T(W) = f(W \cup T) - f(W)$.

---

[*] Corresponding author.

35th Conference on Neural Information Processing Systems (NeurIPS 2021).

On the one hand,

$$\rho_T(W) = f(W \cup T) - f(W) = \mathcal{I}(G + T) - \mathcal{I}(G + W \cup T) = \boldsymbol{s}^\top (\boldsymbol{I} + \boldsymbol{L})_T^{-1} \boldsymbol{s} - \boldsymbol{s}^\top (\boldsymbol{I} + \boldsymbol{L})_{W \cup T}^{-1} \boldsymbol{s}$$

$$= \boldsymbol{s}^\top \left( \sum_{i=1}^{n-1} \frac{1}{1 + \lambda_i(\boldsymbol{L}_T)} \boldsymbol{u}_i \boldsymbol{u}_i^\top - \frac{1}{1 + \lambda_i(\boldsymbol{L}_{W \cup T})} \boldsymbol{u}_i \boldsymbol{u}_i^\top \right) \boldsymbol{s}$$

$$= \boldsymbol{s}^\top \left( \sum_{i=1}^{n-1} \frac{\lambda_i(\boldsymbol{L}_{W \cup T}) - \lambda_i(\boldsymbol{L}_T)}{(1 + \lambda_i(\boldsymbol{L}_T))(1 + \lambda_i(\boldsymbol{L}_{W \cup T}))} \boldsymbol{u}_i \boldsymbol{u}_i^\top \right) \boldsymbol{s} \geq \boldsymbol{s}^\top \left( \frac{\boldsymbol{L}_{W \cup T} - \boldsymbol{L}_T}{(1 + \lambda_{n-1}(\boldsymbol{L}_T))(1 + \lambda_{n-1}(\boldsymbol{L}_{W \cup T}))} \right) \boldsymbol{s}$$

$$= \boldsymbol{s}^\top \left( \frac{\sum_{e \in W \setminus T} \boldsymbol{b}_e \boldsymbol{b}_e^\top}{(1 + \lambda_{n-1}(\boldsymbol{L}_T))(1 + \lambda_{n-1}(\boldsymbol{L}_{W \cup T}))} \right) \boldsymbol{s}.$$

On the other hand,

$$\rho_T(W) = \boldsymbol{s}^\top \left( \sum_{i=1}^{n-1} \frac{\lambda_i(\boldsymbol{L}_{W \cup T}) - \lambda_i(\boldsymbol{L}_T)}{(1 + \lambda_i(\boldsymbol{L}_T))(1 + \lambda_i(\boldsymbol{L}_{W \cup T}))} \boldsymbol{u}_i \boldsymbol{u}_i^\top \right) \boldsymbol{s}$$

$$\leq \boldsymbol{s}^\top \left( \frac{\boldsymbol{L}_{W \cup T} - \boldsymbol{L}_T}{(1 + \lambda_1(\boldsymbol{L}_T))(1 + \lambda_1(\boldsymbol{L}_{W \cup T}))} \right) \boldsymbol{s} = \boldsymbol{s}^\top \left( \frac{\sum_{e \in W \setminus T} \boldsymbol{b}_e \boldsymbol{b}_e^\top}{(1 + \lambda_1(\boldsymbol{L}_T))(1 + \lambda_1(\boldsymbol{L}_{W \cup T}))} \right) \boldsymbol{s}.$$

Putting the above two bounds together leads to

$$\frac{\sum_{e \in W \setminus T} \rho_e(T)}{\rho_T(W)} \geq \boldsymbol{s}^\top \left( \sum_{e \in W \setminus T} \frac{\boldsymbol{b}_e \boldsymbol{b}_e^\top}{(1 + \lambda_{n-1}(\boldsymbol{L}_T))(1 + \lambda_{n-1}(\boldsymbol{L}_{T+e}))} \right) \boldsymbol{s} \times \frac{(1 + \lambda_1(\boldsymbol{L}_T))(1 + \lambda_1(\boldsymbol{L}_{T \cup W}))}{\sum_{e \in W \setminus T} \boldsymbol{s}^\top \boldsymbol{b}_e \boldsymbol{b}_e^\top \boldsymbol{s}}$$

$$\geq \left( \frac{1 + \lambda_1(\boldsymbol{L})}{1 + \lambda_{n-1}(\boldsymbol{L}_{E_C})} \right)^2,$$

which implies the lower bounds of $\gamma$.

Similarly, we derive the upper bound of the curvature $\alpha$. Let $j$ be any candidate edge in $W \setminus T$. Then,

$$\frac{\rho_j(W \setminus j \cup W)}{\rho_j(T \setminus j)} \geq \frac{\boldsymbol{s}^\top \boldsymbol{b}_j \boldsymbol{b}_j^\top \boldsymbol{s}}{(1 + \lambda_{n-1}(\boldsymbol{L}_T))(1 + \lambda_{n-1}(\boldsymbol{L}_{T+e}))} \times \frac{(1 + \lambda_1(\boldsymbol{L}_T))(1 + \lambda_1(\boldsymbol{L}_{T \cup W}))}{\boldsymbol{s}^\top \boldsymbol{b}_j \boldsymbol{b}_j^\top \boldsymbol{s}} \geq \left( \frac{1 + \lambda_1(\boldsymbol{L})}{1 + \lambda_{n-1}(\boldsymbol{L}_{E_C})} \right)^2,$$

which combining with the definition of curvature completes the proof. $\square$

## 4 Proof of Theorem 5.1

**Proof.** To show the non-submodularity of the function concerned, consider the graph in Figure 1, which is a 5-node path-graph with $e_1$ and $e_2$ being inexistent edges. We set initial opinion vector as $\boldsymbol{s} = (0.25, 0.5, 0.5, 0.5, 0.25)^\top$, and define two edge sets, $T = \emptyset$ and $W = \{e_2\}$. Then,

$$\mathcal{I}(T) = 0.8295, \mathcal{I}(T + e_1) = 0.8295,$$
$$\mathcal{I}(W) = 0.8227, \mathcal{I}(W + e_1) = 0.8220,$$

so that

$$\mathcal{I}(T) - \mathcal{I}(T + e_1) = 0 < 0.007 = \mathcal{I}(W) - \mathcal{I}(W + e_1),$$

which violates the definition of submodularity. Thus, it follows that the set function of our problem is non-submodular. $\square$

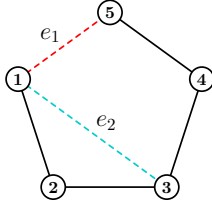

Figure 1: A 5-nodes path-graph where $e_1$ and $e_2$ are inexistent edges.

# 5 Proof of Lemma 6.3

**Proof.** According to the assumption:

$$(1 - \frac{\epsilon}{12})\|\overline{\boldsymbol{X}}\boldsymbol{e}_u\|^2 \le \|\boldsymbol{X}'\boldsymbol{e}_u\|^2 \le (1 + \frac{\epsilon}{12})\|\overline{\boldsymbol{X}}\boldsymbol{e}_u\|^2$$

holds for any node $u \in V$ and

$$(1 - \frac{\epsilon}{12})\|\overline{\boldsymbol{X}}\boldsymbol{b}_e\|^2 \le \|\boldsymbol{X}'\boldsymbol{b}_e\|^2 \le (1 + \frac{\epsilon}{12})\|\overline{\boldsymbol{X}}\boldsymbol{b}_e\|^2$$

holds for any pair of nodes $u$ and $v$ connecting an edge $e$ in $E$.

As $\mathcal{G}$ is connected, there exists a simple path $P_{uv}$ connecting $u$ and $v$. By applying the triangle inequality twice, we obtain

$$|\|\tilde{\boldsymbol{X}}\boldsymbol{b}_e\| - \|\boldsymbol{X}'\boldsymbol{b}_e\|| \le \|(\tilde{\boldsymbol{X}} - \boldsymbol{X}')\boldsymbol{b}_e\| \le \sum_{(a,b)\in P_{uv}} \|(\tilde{\boldsymbol{X}} - \boldsymbol{X}')(\boldsymbol{e}_a - \boldsymbol{e}_b)\|.$$

We will upper bound the last term by considering its square:

$$\left( \sum_{(a,b)\in P_{uv}} \|(\tilde{\boldsymbol{X}} - \boldsymbol{X}')(\boldsymbol{e}_a - \boldsymbol{e}_b)\| \right)^2 \le n \sum_{(a,b)\in P_{uv}} \|(\tilde{\boldsymbol{X}} - \boldsymbol{X}')(\boldsymbol{e}_a - \boldsymbol{e}_b)\|^2 \le n \sum_{(a,b)\in E} \|(\tilde{\boldsymbol{X}} - \boldsymbol{X}')(\boldsymbol{e}_a - \boldsymbol{e}_b)\|^2$$

$$= n\|(\tilde{\boldsymbol{X}} - \boldsymbol{X}')\boldsymbol{B}^\top\|_F^2 = n\|\boldsymbol{B}(\tilde{\boldsymbol{X}} - \boldsymbol{X}')\|_F^2.$$

Note that the first inequality is derived by Cauchy-Schwarz Inequality. Below we transform the above-obtained Frobenius norm $n\|\boldsymbol{B}(\tilde{\boldsymbol{X}} - \boldsymbol{X}')\|_F^2$ into the $(\boldsymbol{I} + \boldsymbol{L})$-norm as

$$n\|\boldsymbol{B}(\tilde{\boldsymbol{X}} - \boldsymbol{X}')\|_F^2 = n\text{Tr}\left((\tilde{\boldsymbol{X}} - \boldsymbol{X}')^\top \boldsymbol{B}^\top \boldsymbol{B}(\tilde{\boldsymbol{X}} - \boldsymbol{X}')\right) = n\text{Tr}\left((\tilde{\boldsymbol{X}} - \boldsymbol{X}')^\top \boldsymbol{L}(\tilde{\boldsymbol{X}} - \boldsymbol{X}')\right)$$

$$\le n\text{Tr}\left((\tilde{\boldsymbol{X}} - \boldsymbol{X}')^\top (\boldsymbol{I} + \boldsymbol{L})(\tilde{\boldsymbol{X}} - \boldsymbol{X}')\right) = n \sum_{i=1}^p (\tilde{\boldsymbol{X}}_i - \boldsymbol{X}'_i)(\boldsymbol{I} + \boldsymbol{L})(\tilde{\boldsymbol{X}}_i - \boldsymbol{X}'_i)^\top$$

$$\le n\delta_1^2 \sum_{i=1}^p \boldsymbol{X}'_i(\boldsymbol{I} + \boldsymbol{L})(\boldsymbol{X}'_i)^\top.$$

Applying the fact that $\boldsymbol{L} \preceq (n+1)\boldsymbol{I}$ and $\boldsymbol{\Omega}\boldsymbol{L}\boldsymbol{\Omega} \preceq \boldsymbol{\Omega} \preceq \boldsymbol{I}$, we have

$$n\delta_1^2 \sum_{i=1}^p \boldsymbol{X}'_i(\boldsymbol{I} + \boldsymbol{L})(\boldsymbol{X}'_i)^\top \le n\delta_1^2(n+1) \sum_{i=1}^p \boldsymbol{X}'_i(\boldsymbol{X}'_i)^\top = n\delta_1^2(n+1)\|\boldsymbol{X}'\|_F^2$$

$$\le n\delta_1^2(n+1) \sum_{i=1}^n (1 + \frac{\epsilon}{12})\|\overline{\boldsymbol{X}}\boldsymbol{e}_i\|^2 \le n\delta_1^2(n+1) \sum_{i=1}^n (1 + \frac{\epsilon}{12})\boldsymbol{e}_i^\top \boldsymbol{\Omega}\boldsymbol{e}_i$$

$$\le n\delta_1^2(n+1)(1 + \frac{\epsilon}{12})n.$$

On the other hand,

$$\|\boldsymbol{X}'\boldsymbol{b}_e\|^2 \ge (1 - \frac{\epsilon}{12})\|\overline{\boldsymbol{X}}\boldsymbol{b}_e\|^2 = (1 - \frac{\epsilon}{12})\boldsymbol{b}_e^\top \boldsymbol{\Omega}\boldsymbol{L}\boldsymbol{\Omega}\boldsymbol{b}_e$$

$$\ge (1 - \frac{\epsilon}{12})\frac{1}{n^2(n+1)^2}\|\boldsymbol{b}_e\|^2 = 2(1 - \frac{\epsilon}{12})\frac{1}{n^2(n+1)^2}.$$

The last inequality is obtained for the following reason. Note that $\boldsymbol{b}_e$ is orthogonal to all-one vector $\boldsymbol{1}$, an eigenvector of $\boldsymbol{L}$ associated with the unique eigenvalue 0. Therefore, $\boldsymbol{b}_e^\top \boldsymbol{L}\boldsymbol{b}_e \ge \lambda_{\min}\|\boldsymbol{b}_e\|^2$ holds. In addtion, $\boldsymbol{L}$ and $(\boldsymbol{I} + \boldsymbol{L})^{-1}$ share identical eigenspaces.

Combining the above-obtained results, it follows that

$$\frac{\left|\|\tilde{\boldsymbol{X}}\boldsymbol{b}_e\| - \|\boldsymbol{X}'\boldsymbol{b}_e\|\right|}{\|\boldsymbol{X}'\boldsymbol{b}_e\|} \le \frac{\delta_1 n(n+1)\sqrt{(1 + \epsilon/12)(n+1)}}{\sqrt{2(1 - \epsilon/12)}} \le \frac{\epsilon}{32},$$

based on which we further obtain

$$\left|\|\tilde{\boldsymbol{X}}\boldsymbol{b}_e\|^2 - \|\boldsymbol{X}'\boldsymbol{b}_e\|^2\right| = \left|\|\tilde{\boldsymbol{X}}\boldsymbol{b}_e\| - \|\boldsymbol{X}'\boldsymbol{b}_e\|\right| \times \left|\|\tilde{\boldsymbol{X}}\boldsymbol{b}_e\| + \|\boldsymbol{X}'\boldsymbol{b}_e\|\right|$$

$$\le \frac{\epsilon}{32}(2 + \frac{\epsilon}{32})\|\boldsymbol{X}'\boldsymbol{b}_e\|^2 \le \frac{\epsilon}{12}\|\boldsymbol{X}'\boldsymbol{b}_e\|^2,$$

which completes the proof. $\square$

# 6 Proof of Lemma 6.5

**Proof.** Since $L$ is the Laplacian of a connected graph, we can find a path $P_{uv}$ connecting $u$ and $v$. By applying the triangle inequality, we obtain

$$q^\top b_e^\top b_e q = (q_u - q_v)^2 \leq n \sum_{(a,b) \in P_{uv}} (q(e_a - e_b))^2$$

$$\leq n \sum_{(a,b) \in E} \|q(e_a - e_b)\| \leq n q^\top L q,$$

which implies that

$$\|q\|_{b_e b_e^\top} \leq \sqrt{n} \|q\|_L.$$

We first bound the value $\left| \|q\|_{b_e b_e^\top} - \|\Omega s\|_{b_e b_e^\top} \right|$ by the triangle inequality

$$\left| \|q\|_{b_e b_e^\top} - \|\Omega s\|_{b_e b_e^\top} \right| \leq \|q - \Omega s\|_{b_e b_e^\top} \leq \sqrt{n} \|q - \Omega s\|_L$$

$$\leq \sqrt{n} \delta_3 \|\Omega s\|_{I+L} = \sqrt{n} \delta_3 \sqrt{s^\top \Omega s}$$

$$\leq \delta_3 \sqrt{n} \sqrt{s^\top s} \qquad \text{since } \|s\|^2 \leq n,$$

$$\leq \delta_3 n,$$

based on which we proceed to bound $\left| \|q\|_{b_e b_e^\top}^2 - \|\Omega s\|_{b_e b_e^\top}^2 \right|$:

$$\left| \|q\|_{b_e b_e^\top}^2 - \|\Omega s\|_{b_e b_e^\top}^2 \right| = \left| \|q\|_{b_e b_e^\top} + \|\Omega s\|_{b_e b_e^\top} \right| \times \left| \|q\|_{b_e b_e^\top} - \|\Omega s\|_{b_e b_e^\top} \right|$$

$$\leq \left( 2\|\Omega s\|_{b_e b_e^\top} + \delta_3 n \right) \delta_3 n \leq \left( 2\sqrt{n} \|\Omega s\|_L + \delta_3 n \right) \delta_3 n$$

$$\leq (2n + \delta_3 n) \delta_3 n \qquad \text{since } \|z\|^2 \leq n, \delta_3 \leq 1 \text{ and } \Omega \leq L^\dagger,$$

$$\leq 3 \delta_3 n^2.$$

Thus, one has

$$\left| \|q\|_{b_e b_e^\top}^2 - \|\Omega s\|_{b_e b_e^\top}^2 \right| \leq 3 \delta_3 n^2 \leq \frac{\epsilon}{3},$$

which leads to the results directly. $\square$

## 6.1 Proof of Theorem 6.1

**Proof.** Using Lemmas $6.3, 6.4$, and $6.5$, one has

$$|\hat{f}(e) - f(e)| = \left| \frac{\|q\|_{b_e b_e^\top}^2}{1 + \|\tilde{X} b_e\|^2 + \|\tilde{Y} b_e\|^2} - \frac{\|\Omega s\|_{b_e b_e^\top}^2}{1 + b_e^\top \Omega b_e} \right|$$

$$\leq \left| \frac{1}{1 - \epsilon/3} \frac{\|q\|_{b_e b_e^\top}^2}{1 + b_e^\top \Omega b_e} - \frac{\|\Omega s\|_{b_e b_e^\top}^2}{1 + b_e^\top \Omega b_e} \right|$$

$$\leq \left| \frac{1}{1 - \epsilon/3} \frac{\|\Omega s\|_{b_e b_e^\top}^2 + \epsilon/3}{1 + b_e^\top \Omega b_e} - \frac{\|\Omega s\|_{b_e b_e^\top}^2}{1 + b_e^\top \Omega b_e} \right|$$

$$= \left| \frac{1}{1 - \epsilon/3} \frac{(z_u - z_v)^2 + \epsilon/3}{1 + r_{uv}} - \frac{(z_u - z_v)^2}{1 + r_{uv}} \right|$$

$$\leq \frac{2\epsilon/3}{1 - \epsilon/3} \leq \frac{4}{5}\epsilon, \qquad \text{since } (z_u - z_v)^2 \leq 1 \text{ and } 0 \leq r_{uv} \leq 2,$$

which leads to the result. $\square$

# References

[1] Carl D Meyer, Jr. Generalized inversion of modified matrices. *SIAM Journal on Applied Mathematics*, 24(3):315–323, 1973.