# OpenReview forum: "Minimizing Polarization and Disagreement in Social Networks via Link Recommendation"
_NeurIPS.cc/2021/Conference — NeurIPS 2021 Poster_

### Official Review · Reviewer_Ceon · 2021-07-13

**Rating:** 6
**Confidence:** 5

**Summary:**

The paper studies the problem of recommending k links to a given graph G from a candidate pool of candidate links Ec. The goal is to minimize the index of disagreement-polarization as introduced by Musco et al. The authors prove that the objective is not submodular, but they analyze the performance of greedy, and provide a fast version based on random projections. They apply their methods on real-world data, with synthetic opinions.

**Limitations And Societal Impact:**

The work could potentially have a positive impact on mitigating issues from  echo chambers, and filter bubbles.

**Main Review:**

In this work the authors focus on recommending k links in order to minimize polarization and disagreement in social networks. Intuitively, in a recommender system links may connect nodes with similar opinions and thus have low disagreement but incur higher polarization, or nodes with totally different opinions and thus have low polarization but incur higher disagreement. The objective takes into account both terms, and aims to balance between the two.  Specifically, the authors rely on the framework of Musco et al. that assumes the Friedkin-Johnsen model of opinion dynamics and defines disagreement as the sum of the squared differences of opinions over all edges, whereas the polarization as the second moment of the opinions centered around 0. Both terms can be nicely expressed using the Laplacian. The authors consider the problem of   minimizing the sum of those terms over all possible recommendations/additions of k edges. The objective is monotone, a statement proved directly from applying the Sherman-Morisson formula.  Clearly, there is a straight-forward naive algorithm that checks all possible subsets of cardinality k among the set of possible recommendations Ec. This is infeasible in practice, and thus the authors study the performance of greedy. The authors prove that the resulting optimization problem is neither submodular nor supermodular, so the greedy algorithm is not guaranteed to provide an 1-1/e approximation ratio or an exact solution. They use the results of [5] to characterize the parameters of how “far” is the objective from being submodular, and provide some insights on how the greedy algorithm behaves; it depends on the graph G, and the graph formed by the possible recommendations that in principal can be the complement of G. The authors also leverage the ideas of Spielman-Srivastava from their work on spectral sparsifiers  that use random projections for fast effective resistance computations in order to speed up the greedy algorithm. Finally they implement their algorithm in Julia (code available in supplement), and test it on a variety of datasets.

Some questions to the authors:

1. Unfortunately the approximation guarantees are not easy to interpret. Perhaps you can use some classes of graphs with known spectral guarantees to derive some interesting corollaries (e.g., expander graphs, stochastic blockmodel).

2. Unfortunately none of the datasets contains some true opinion dynamics. Therefore it would have been more interesting to see more distributions other than the uniform.

3. Did you consider the case of directed graphs? The framework of Musco et al. does not seem to be directly applicable there.


====
Post-rebuttal: I thank to the authors for their detailed response. I have upgraded my score.

**Time Spent Reviewing:**

4

---

> ### Author Response · Authors · 2021-08-10
> **Response to Reviewer4**
>
> Thanks for your time and help.
>
> $\textbf{Approximation guarantee}$: Our greedy algorithm has a tight approximation guarantee of $\frac{1}{\alpha}(1-e^{-\alpha\gamma})$ for non-submodular functions [5]. For the objective function of our problem, we derive the bounds for $\alpha$ and $\gamma$, which together with their proofs are provided in Supplementary Material. The proofs are primarily based on the fact that the non-zero eigenvalues of Laplacian matrix of a graph will increase when adding edges to the graph. We first write $\mathbf{L}$ as an eigendecomposition form: $\mathbf{L}= \sum_{i=1}^{n-1} \lambda_i  \mathbf{u_i}  \mathbf{u_i}^{\top}$. Then it follows that $\lambda_i(\mathbf{L}) \leq \lambda_i(\mathbf{L_T})\leq \lambda_i(\mathbf{L_{E_C}})$. Then one obtains the bounds for $\alpha$ and $\gamma$. We hope these analyses are helpful for understanding the approximation guarantee you are concerned with.
>
>
> $\textbf{Real-world networks with true opinion}$: By definition, the computation of disagreement and polarization involves the initial opinion and expressed opinion. By applying opinion mining and sentiment analysis techniques [Liu Bing, Sentiment Analysis: mining sentiments, opinions, and emotions, 2015, Cambridge University Press Cambridge], the expressed opinion of an agent is readily observable in a social network. However, the innate opinions of agents are not accessible, which are often hidden. Although Das et al. [17] have proposed a near-optimal sampling algorithm for estimating the true average innate opinion of social networks, their algorithm cannot be used to evaluate the innate opinion of every node. Thus, it is difficult to evaluate disagreement and polarization based on the actual available data. We turn to specific opinion distributions.
>
> $\textbf{Opinion distributions of $\mathbf{s}$}$: From related lemmas, theorems, as well as their proofs, one can see that our algorithms and their performance are independent of selection of initial opinions. We thus only showed the experimental results for uniform distribution of initial opinions. It is expected that for other distributions of initial opinions, such as exponential distribution and power-law distribution, the results are similar to those corresponding to uniform distribution displayed in Figures 1 and 2, which are omitted due to the space limitations. Anyway, we are very grateful to you for your valuable suggestion, we will add the experimental results for other initial opinions to the final version, if our paper was accepted.
>
>
> $\textbf{Directed graphs}$: Since our considered problem is based on the framework of Musco et al [36], our proposed algorithm does not work on directed graphs. In future, it would be of great interest to extend the definition of disagreement and polarization to directed networks and modify our algorithm on directed graphs.

---

### Official Review · Reviewer_HjPc · 2021-07-15

**Rating:** 7
**Confidence:** 4

**Summary:**

In this paper, the authors study an opinion dynamics problem using the Friedkin-Johnsen model, where one aims to minimize the polarization + disagreement by adding up to k edges. The authors derive a greedy algorithm for this problem and show that it has a constant factor approximation. They then derive a faster version of the greedy algorithm (but without a constant factor approximation) involving random matrix projections, JL, and fast SDDM solvers. Empirically, they show that their methods are close to optimal for small k, and that their fast greedy algorithm scales well to large networks.

**Limitations And Societal Impact:**

No, the potential negative societal impact of their work is not discussed. I would add at least one example of how this could be used negatively (eg manipulation by social networks)

**Main Review:**

Strengths:
* Novel+interesting opinion dynamics problem
* Clean analysis of algorithms
* Empirical results are very good
* Writing is good for the most part

Weaknesses:
* Writing in Section 6 could be clearer
* Could use better experimental baselines

Overall, I liked this paper and would like to see it accepted. The problem itself seems novel and interesting - similar to [36], but completely different type of analysis. The greedy algorithms flow quite naturally from the observation in Lemma 4.1, that P+D index goes down as you add more edges. (On a side note, this observation was not immediately obvious to me, e.g. if you have a barbell-like graph, with two dense clusters that are not that connected to each other, and you add edges between nodes in the same cluster - I would guess polarization goes up and disagreement goes down, but unclear what happens to P+D). The empirical results are also very impressive, showing that your fast algorithm can run on graphs with millions of nodes/edges.

I only have minor complaints; see below.

=======

* Section 6: I find the X, X-bar, X-tilde notation (line 252) to be quite confusing. Seems easier to just write these formulas out directly without introducing X’s. In general this excess of notation makes Section 6 hard to read.

* It should be made explicit that Algorithm 2 does not have any guarantees (eg constant factor approx) unlike Algorithm 1. The writing is a bit too slick, making it seem like Algorithm 2 has some global guarantees (eg by mentioning \epsilon in the runtime) when it does not.

* I find the baselines in the experiments a bit lacking since none of them use the innate opinions S. It would be good to add more baselines that use s.

* Question: Do any results extend to convex combinations of polarization + disagreement? E.g. [36] notes that convex combinations P + rho*D of polarization and disagreement are generally not convex (except for when rho=1). I wonder if that affects Lemma 4.1 at all.

* Line 221: I understand space is limited but it would be interesting to see the approximation guarantee for the naive greedy algorithm written out, since you say in line 223 that it is not satisfactory

* Line 8 of SpGreedy: I find the interchanging of (I+L)^{-1} and \Omega to be confusing notation since they are the same thing.  In general this is confusing notation throughout the paper.

* Line 233: would be good to define what SDDM means

* Table 1: What is the k (number of edges added) here? Should be included somewhere, eg in table caption

**Nits/typos**:
* line 41: to greedy -> to a greedy
* line 161: “in the sequel” unsure what this means
* line 172: Similar idea -> A similar idea
* Supplement 1: Should say proof of 4.1, not 3.1
* Problem 1: I assume E and E_C are disjoint?
* Line 191: “by the following” -> “in the following”

* Line 8 of supplement: capitalize “when”
* Line 26 of supplement: L_{W + T} should be L_{W \cup T}
* Line 27 of supplement: how did you get the last inequality?

**Time Spent Reviewing:**

5.5

---

> ### Author Response · Authors · 2021-08-10
> **Response to Reviewer3**
>
> Thank you for your time and insightful comments on our paper. We are encouraged that you found our problem novel and interesting. Below is a summary of our response to your comments.
>
> $\textbf{Notation}$: To make the paper more concise, we will reduce unnecessary notation, such as $\mathbf{X}$, $\mathbf{\bar{X}}$, $\mathbf{\tilde{X}}$ in line 252.
>
> $\textbf{Guarantee of Algorithm 2}$: We will state clearly that unlike Algorithm 1, Algorithm 2 does not have any guarantees.
>
> $\textbf{Baselines in Experiments}$: According to related lemmas, theorems, and their proofs, one can observe that our algorithms and their performance are independent of the initial opinions. Thus, in our Experiments Section, we only showed the results for uniform distribution of initial opinions. It is expected that for other distributions of initial opinions, such as exponential distribution and power-law distribution, the results are similar to those corresponding to uniform distribution displayed in Figures 1 and 2, which are omitted due to the space limitations. Anyway, thank you for this valuable suggestion, we would add more experimental results for other baselines related to innate opinions.
>
> $\textbf{Convex combination of polarization and disagreement}$: As shown in [36], for the convex combinations $P + \rho \times D$ of polarization and disagreement, they are generally not convex. These combinations significantly affect our results. For example, Lemma 4.1 dose not hold surely other than $\rho=1$.
>
> $\textbf{Approximation guarantee}$: We will provide the approximation guarantee according to Lemma 5.1.
>
> $\textbf{Confusing notation in Algorithm SpGreedy}$: For simplicity, we will replace $(I+L)^{-1}$ in SpGreedy by $\Omega$. Thank you.
>
> $\textbf{Meaning of SDDM in line 233}$: We will provide the meaning for SDDM. Specifically, SDDM indicates symmetric, diagonally-dominant M-matrix.
>
> $\textbf{Table 1}$: In the caption of Table 1, we will state clearly that the running time and results correspond to $k=50$.
>
> $\textbf{Sets $E$ and $E_C$ in Problem 1}$: Yes, as we mention in Problem 1, $E_C$ is the set of nonexistent edges, while $E$ is the set of all existing edges. Thus, $E$ and $E_C$ are disjoint.
>
> $\textbf{Explanation for Line 27 in supplementary material}$: The last inequality is obtained for the following reasons. The non-zero eigenvalues of the Laplacian matrix of a graph will increase when adding edges to the graph, thus, both the term $\lambda_1(\mathbf{L_T})$ and the term $\lambda_1(\mathbf{L_{T\cup W}})$ is larger than $\lambda_1(\mathbf{L})$. Similarly, both the term $\lambda_{n-1}(\mathbf{L_T})$ and the term $\lambda_{n-1}(\mathbf{L_{T\cup W}})$ is smaller than $\lambda_{n-1}(\mathbf{L_{E_C}})$.
>
> $\textbf{Nits/typos}$: Thank you for your careful reading. We will check carefully the whole paper and Supplementary Material, correcting all typos.
>
> $\textbf{Ethical concerns}$: The quick growth or development of online social networks and AI has changed the way people interact and make decisions. The extent of this influence could be observed not only in marketing and social behavior but also in referendums and elections, leading to distortion of democratic manifestations and representations. In our paper, based on the framework of Musco et al [36], we consider problem of minimizing disagreement and polarization from the theoretical angle, which involves both the initial opinion and expressed opinion. Since both opinions are not directly accessible, our work incurs little ethical issue at present, as in the work of Musco et al [36]. However, since the consequences of the misuse of Internet based social networks could have a potential negative impact on society, it is important to define ethical guidelines and policies for developers, rulers, operators, and social actors. As researchers, we will definitely abide by these guidelines and policies.

---

> > ### Comment · Reviewer_HjPc · 2021-08-11
> > **Thanks for your response.**
> >
> > Thanks for your thorough response. I am satisfied with your responses to my questions.
> >
> >  While I agree with the other critiques raised by other reviewers - in particular, I agree that the experiments should consider other (non-uniform) distributions of initial opinions s - I still quite like this paper and would like to see it accepted. Thus, my score of "accept" remains unchanged.

---

> > > ### Author Response · Authors · 2021-09-01
> > > **Response to Reviewer HjPc**
> > >
> > > We have carefully considered the suggestion of all reviewers and we will add more experimental results in terms of other distribution of initial opinions $s$. We sincerely thank you again for your  support and constructive comments on our paper.

---

### Official Review · Reviewer_nZrq · 2021-07-16

**Rating:** 6
**Confidence:** 3

**Summary:**

In this paper the authors propose the task of adding links to a social network to minimize notions of disagreement and polarization across the network.  In particular, they define disagreement across neighbors in the graph and polarization by the variance in expressed opinions of the nodes.  The paper then offers two approximate algorithms for greedily adding a fixed number of edges to the graph to minimize the sum of the disagreement and polarization.  They show that their algorithms are both theoretically and empirically tractable, outperform naive baselines, and in some cases close to optimal.


**Ethical Concerns:**

The framing of polarization as defined here and making that a goal of social network design / recommendation raises some ethical concerns.  In particular, the framing seems to be one that is suitable for theoretical analysis, but it is not clear that merely decreasing variance of opinions is an ethically good definition of polarization to optimize or if networks should be taking such actions.  A more complete discussion of these limitations beyond the scope of the theory would be valuable.


**Limitations And Societal Impact:**

See above and below.


**Main Review:**

Overall the paper does a nice job of clearly providing their problem formulation and demonstrating their ability to provide reasonably fast, approximate algorithms to solve these problems.

My primary concern with the paper is with respect to its significance.  While I believe it is theoretically interesting, I worry that it is overly simplified of a model in ignoring any way of incorporating user preferences or a traditional recommender providing probability of the users being interested in connecting, as past work has done.  A discussion of how the research could be incorporated into more practical settings would be valuable.  That said, the theoretical result is still interesting and this may be unnecessary even if valuable.

Details:

Eval - initializing with uniformly random $s$ is particularly unrealistic, and I'd be curious how results are effected by other choices of $s$, e.g. correlation opinions with graph structure as in communities.

None of the baselines consider the opinions $s$?  They do show the effect of merely making the graph more dense but I'd think connecting individuals or clusters with very different opinions is critical.


**Time Spent Reviewing:**

2.5

---

> ### Author Response · Authors · 2021-08-10
> **Response to Reviewer2**
>
> Thank you for your time and help.
>
> As in most previous work, we adopt the Friedkin-Johnsen (FJ) model to study optimization and algorithms for polarization and disagreement. Despite its simple but succinct nature, the FJ model incorporates French's ''theory of social power'' [French Jr, J. R. A formal theory of social power. Psychological Review, 63, 181–194, 1956.], sufficiently grasping complicated social behaviors. Moreover, several empirically studies indicated that the FJ model captures to a large extent some mechanisms for opinion formation. The following papers are two examples.
>
> (1) A theory of the evolution of social power: Natural trajectories of interpersonal influence systems along issue sequences. Sociol Sci, 2016, 3: 444–472.
> (2) How truth wins in opinion dynamics along issue sequences, PNAS, 2017, 114: 11380–11385.
>
> Admittedly, it would be of great interest to incorporate user preferences or a traditional recommender providing probability of the users being interested in connecting. Here, we focus on the optimization and algorithms of polarization and disagreement by the operation of edge addition, which is a practical approach of graph edit, different from traditional recommending algorithm providing probability. We expect your understanding and consideration.
>
>
> $\textbf{Initial opinions $\mathbf{s}$}$: From the analysis of lemmas, theorems, as well as their proofs, one can see that our proofs, algorithms and their performance do not depend on the distributions of initial opinions. In our experiments, we only showed the results uniformly random distribution of the initial opinions, due to the space limitations. It is expected that for other distributions of initial opinions, such as exponential distribution and power-law distribution, the results are similar to those corresponding to uniform distribution displayed in Figures 1 and 2, which are omitted due to the space limitations. Anyway, we are very grateful to you for your constructive suggestion, we will add the experimental results for other initial opinions to the final version, if our paper was accepted.
>
> $\textbf{Baselines considering initial opinions $\mathbf{s}$}$: In addition, we have also considered $\mathbf{s}$-related baseline. However, we did not adopt this baseline in our paper for the following reason. According to the results in Lemma 4.1 in line 164, the variation of the P-D index under the perturbation of a single edge connecting nodes $u$ and $v$ depends on two terms: $(\mathbf{z_u-z_v})^2$ and forest distance $r_{u, v}$, where $\mathbf{z_u}$ and forest diatance are in fact dependent on the topology of the network. Thus, simply considering initial opinion $\mathbf{s}$ is not sufficient to find optimal edges. For example, it is intuitive that connecting two individuals with large difference of initial opinions may lead to the large decrease of the P-D index. Actually, this is not the case. Let us consider a simple example, a 4-node path graph $G$ with initial opinion vector $\mathbf{s}= (0.78, 0.3, 0.45, 0.43)^\top$. Let $e_{1,3}$ and $e_{1,4}$ be the nonexistent edges connecting node pairs $(1,3)$ and $(1,4)$, respectively. By Lemma 4.1, the adding of $e_{1,3}$ results in a decrease of the P-D index by $f(e_{1,3})=0.015$, while creating $e_{1,4}$ yields a decrease of the P-D index by $f(e_{1,4})=0.014$, smaller than $0.015$, which is counter-intuitive. Thus, our methods can find optimal edges, comparing with this baseline. Anyway, thank you very much for your valuable suggestion. We will add some experimental results for $\mathbf{s}$-related baseline to the final version to better demonstrate our performance, if our paper was accepted.
>
>
> $\textbf{Ethical concerns}$: The quick growth or development of online social networks and AI has changed the way people interact and make decisions. The extent of this influence could be observed not only in marketing and social behavior but also in referendums and elections, leading to distortion of democratic manifestations and representations. In our paper, based on the framework of Musco et al [36], we consider problem of minimizing disagreement and polarization from the theoretical angle, which involves both the initial opinion and expressed opinion. Since both opinions are not directly accessible, our work incurs little ethical issue at present, as in the work of Musco et al [36]. However, since the consequences of the misuse of Internet based social networks could have a potential negative impact on society, it is important to define ethical guidelines and policies for developers, rulers, operators, and social actors. As researchers, we will definitely abide by these guidelines and policies.

---

> > ### Comment · Reviewer_nZrq · 2021-08-31
> > **Response**
> >
> > Thank you for taking time to respond to my concerns.  A few responses:
> > - On baselines: I appreciate the intuition but also feel that having more baselines with equal information will strengthen the paper. I appreciate you incorporating this in your next draft.
> > - On ethical concerns: I agree the paper is exploring an important space.  That said, as the authors motivate the work and its novelty by social concerns such as "filter bubbles," expanded exposition on the design choices and their implications I think could still be valuable.

---

> > > ### Author Response · Authors · 2021-09-01
> > > **Response to Reviewer nZrq**
> > >
> > > We sincerely thank you for providing very helpful comments to guide our revision. We will add experimental results for more baselines to improve the quality of our manuscript. In addition, we feel encouraged for your appreciation on the value of our paper. Thanks again for your time.

---

### Official Review · Reviewer_9U5a · 2021-07-17

**Rating:** 7
**Confidence:** 3

**Summary:**

The problem considered is adding links to a network to decrease the polarization+disagreement ratio. The proposed method is the greedy algorithm: adding links one-by-one to maximize the improvement. Showing approximation ratio based on the approximation result of [5] (in ICML'17, showing that the  polarization+disagreement  is close enough to being submodular.

Evaluating a greedy step exactly required a matrix inversion, hence it is slow. The authors also offer a faster approximate solution using Johnson-Lindenstrauss lemma

**Limitations And Societal Impact:**

Not aware of negative societal impact

**Main Review:**

I find the result showing that  polarization+disagreement  is close enough to being submodular to give guarantees for approximation interesting! I am a bit disappointed that all these details were pushed to supplemental material, including all proofs as well as even including the definition of curvature and submodularity ratio.

The faster approximate version is important to make the method practical.

I am not sure if NeurIPS is the right venue for this work, but it is definitely very interesting

**Time Spent Reviewing:**

1 hour

---

> ### Author Response · Authors · 2021-08-10
> **Response to Reviewer1**
>
>
> Thank you for your time and help.
>
> We are encouraged you found our analysis about guarantees for approximation interesting. If the paper was accepted, we will move the part about guarantees for approximation from the supplementary material to the main text.
>
> We think that our paper falls within the scope of NeurIPS. The main justifications are as follows. On one hand, the subject of our paper belongs to computational social science, which is an important branch of AI and machine learning. The combination of online social networks, portable devices, and AI has changed the way people interact and make decisions. On the other hand, our paper is also related to optimization and algorithms of polarization and disagreement, as measures of two important social phenomena. In very recent years, many papers about optimization and algorithms for relevant problems in social networks have appeared in NeurIPS. For example, Ref. [43] studied co-exposure maximization problem in online social networks, while Ref. [22] addressed the problem of balancing the information exposure in social networks. Ref. [43] and Ref. [22] were published in the proceedings of NeurIPS of 2020 and 2017, respectively.

---

### Decision · Program_Chairs · 2021-09-27

**Decision:**

Accept (Poster)

**Comment:**

Generally all reviews for this paper were positive -- they appreciated the algorithmic contribution -- showing that a greedy approach to link recommendation for minimizing polarization+disagreement gives a bounded approximation ratio, despite the fact that the polarization+disagreement is not submodular. There were significant concerns about practical impact/relevance of the work. Nevertheless, I am recommending acceptance.